# Spatiotemporal Characterization of Dengue Incidence and Its Correlation to Climate Parameters in Indonesia

**DOI:** 10.3390/insects15050366

**Published:** 2024-05-17

**Authors:** Yonny Koesmaryono, Ardhasena Sopaheluwakan, Rini Hidayati, Bambang Dwi Dasanto, Rita Aryati

**Affiliations:** 1Center for Applied Climate Information and Services, Indonesian Agency for Meteorology Climatology and Geophysics, Jakarta 10720, Indonesia; mamenun@bmkg.go.id; 2Department of Geophysics and Meteorology, IPB University, Bogor 16680, Indonesia; rinihidayatigfm@gmail.com (R.H.); bambangdwi@apps.ipb.ac.id (B.D.D.); 3Deputy for Climatology, Indonesian Agency for Meteorology Climatology and Geophysics, Jakarta 10720, Indonesia; ardhasena@bmkg.go.id; 4Center for Climate Risk and Opportunity Management in South Asia Pacific, IPB University, Bogor 16143, Indonesia; 5Directorate of Prevention and Control of Infectious Diseases, Ministry of Health, Jakarta 12950, Indonesia; aryati_rita@yahoo.co.id

**Keywords:** dengue, incidence rate, climate, hotspot area, Indonesia

## Abstract

**Simple Summary:**

Dengue is one of the vector-borne diseases spreading to all provinces in Indonesia. In this study, dengue hotspot areas were identified using a clustering approach and the climate–dengue spatiotemporal distribution and their relations were analyzed using the Singular Value Decomposition technique. Four clusters of dengue hotspot areas were identified. Cluster 1 comprised cities with medium to high incidence rates and high case densities in a narrow area. Cluster 2 has high incidence rates but low case densities, and clusters 3 and 4 featured medium and low incidence rates and case densities, respectively. This dengue clustering can be utilized to determine areas to prioritize for the prevention and control of dengue, i.e., clusters 1 and 2 are high-priority areas, cluster 3 is a medium-priority area, and cluster 4 is a low-priority area. Investigation of the dominant climate variables showed that relative humidity and rainfall were the most influential parameters on incidence rate across all clusters. Temporal fluctuations in the first mode of incidence rate and climate parameters were delineated. The spatial distribution of heterogeneous correlation between the first mode of rainfall and relative humidity to incidence rate exhibited higher values, which were predominantly found in the southern part of Indonesia.

**Abstract:**

Dengue has become a public health concern in Indonesia since it was first found in 1968. This study aims to determine dengue hotspot areas and analyze the spatiotemporal distribution of dengue and its association with dominant climate parameters nationally. Monthly data for dengue and climate observations (i.e., rainfall, relative humidity, average, maximum, and minimum temperature) at the regency/city level were utilized. Dengue hotspot areas were determined through K-means clustering, while Singular Value Decomposition (SVD) determined dominant climate parameters and their spatiotemporal distribution. Results revealed four clusters: Cluster 1 comprised cities with medium to high Incidence Rates (IR) and high Case Densities (CD) in a narrow area. Cluster 2 has a high IR and low CD, and clusters 3 and 4 featured medium and low IR and CD, respectively. SVD analysis indicated that relative humidity and rainfall were the most influential parameters on IR across all clusters. Temporal fluctuations in the first mode of IR and climate parameters were clearly delineated. The spatial distribution of heterogeneous correlation between the first mode of rainfall and relative humidity to IR exhibited higher values, which were predominantly observed in Java, Bali, Nusa Tenggara, the eastern part of Sumatra, the southern part of Kalimantan, and several locations in Sulawesi.

## 1. Introduction

Dengue is a viral disease transmitted by mosquitoes, is prevalent in tropical and subtropical regions, and poses a significant risk to human life. Globally, the World Health Organization (WHO) reported that over the last five decades, the number of dengue cases has increased 30 times [1,2]. An increase in new infection dengue cases occurs yearly, reaching around 50–100 million, with 20,000 deaths in more than 100 dengue-endemic countries, namely Southeast Asia, Africa, America, the eastern Mediterranean, and the Western Pacific, including Australia [1]. According to a WHO report in 2012, from 2004 to 2010, Indonesia was the country with the second-highest number of dengue cases (129,435 cases). In Indonesia, dengue was first found in Surabaya (East Java) and Jakarta in 1968 and then became a public health problem. Dengue has been present in all provinces in Indonesia since 2014, and in 2019, dengue spread to 481 of the 514 regencies/cities. Over the past 50 years, dengue incidence has significantly risen in Indonesia, which in 2009 and 2016, recorded the most significant number of cases [3]. In contrast, the annual Case Fatality Rate (CFR) of dengue has significantly decreased over time, from more than 20% of those infected in the late 1960s to 0.79% in 2016 [3]. 

Dengue is one of 17 Neglected Tropical Diseases (NTDs) that are influenced by vectors (mosquitoes), agents (viruses), hosts (humans), and the environment. Dengue is transmitted through the bite of female *Aedes aegypti* infected with one of the four serotypes of the dengue virus, namely DEN-1, DEN-2, DEN-3, and DEN-4 [1,4]. Human factors, such as age, population density, socioeconomic, human behavior, and mobility, can influence dengue transmission [5,6]. The availability of natural and non-natural mosquito breeding places, often related to human activities, is also crucial in the transmission and spreading of dengue. The development of urban areas, deforestation, and human mobility through traveling, urbanization, and trading can lead to an increase in dengue transmission [6,7], even in non-endemic regions in which no previous dengue cases have been found [8]. Environmental factors such as climate [2,9,10,11,12] and water conditions can affect the life cycle of the mosquito. 

Climate has a significant role in providing a suitable environment for the life cycle of *Aedes aegypti*, from the egg to adult mosquitoes [13]. Rainfall, temperature, and humidity are the climatic factors that influence the development of *Aedes aegypti* [5,14,15]. Naturally, rainfall will fill water reservoirs, which become breeding places for *Aedes aegypti* during egg-laying, larvae, and pupae stages [16,17,18]. The availability of sufficient water as a habitat for mosquito development in both natural and non-natural habitats, such as used tires, pots, and other water storage containers, due to human activities, can increase dengue incidence [10]. Increasing temperature affects the development of viruses and mosquito larvae, vector survival ability, gonotrophic cycle, incubation period, and viral replication [19,20]. Air humidity provides favorable environmental conditions for mosquitoes to survive in their resting places and breeding sites [16]. Climate indirectly influences dengue virus transmission through the possibility of interaction between mosquito vectors and humans [5]. 

Numerous studies have investigated the influence of both climate and non-climatic factors on dengue incidence, including dengue model prediction in Indonesia [21]. On a daily scale, rainfall of 50 mm and humidity of 70% with a time lag of 50 days showed a positive correlation and increase in dengue cases in Semarang [22]. Model prediction of dengue using rainfall showed a positive correlation to dengue incidence in Indramayu, Bogor, North Jakarta, and Padang [23]. Rainfall and humidity in Jakarta also correlate to dengue incidence with values of 0.55 and 0.78, respectively [24]. The optimal daily temperature for dengue infection ranges from 24.3 °C to 30.5 °C [22]. A positive correlation and low magnitude were obtained between the difference in maximum and minimum temperature to dengue incidence in Sleman [25]. Predicting dengue occurrences based on climate can be a preventive measure in dengue control. However, current developments indicate that research on the relationship between dengue and climate in Indonesia remains sporadic and only in a few locations. A comprehensive nationwide study to identify the dominant climate parameters affecting dengue fever incidence has yet to be conducted [21]. At the national level in Indonesia, identifying dengue hotspot areas is crucial for determining the highest priority regions for dengue control. A similar study of macroscale analyses for spatiotemporal patterns of dengue has been conducted in the State of Mato Grosso, Brazil [26]. 

This study aims to determine the hotspot areas of dengue and analyze the nationwide spatiotemporal distribution of dengue incidence associated with dominant climate parameters. The findings of this study can be utilized to identify priority areas for the prevention and control of dengue in the public health system of Indonesia, to provide insights into climate-influenced dengue occurrences at a national level, and as input in developing dengue prediction models using climate parameters as early warning information.

## 2. Methodology

### 2.1. Study Area

Indonesia is a large tropical country located between the Pacific Ocean and the Indian Ocean (6° N–11° S and 95° E–141° E) with over 17,000 islands. With only around 7000 islands inhabited, it has several major islands: Sumatra, Kalimantan, Java, Bali, Nusa Tenggara, Sulawesi, Maluku, and Papua. Indonesia is the fourth most populated country in the world, with a total population of over 270 million people and an area of around 1,892,410 km^2^. In 2022, Indonesia had 38 provinces with 416 regencies and 98 cities. Indonesia has a tropical climate with three dominant types of rainfall pattern, namely (1) Region A: monsoon type with one peak in the wet season in November to March, (2) Region B: Equatorial type with two peaks in the wet season in October to November and March to May, and (3) Region C: local type with one peak in the wet season in June to July [27]. In lowland areas, the average temperature normally ranges from 23.3 °C to 31.6 °C, and in highland areas, the range of average temperature is from 17.6 °C to 27.0 °C [28].

### 2.2. Dengue Data

Dengue case data were obtained from the Directorate of Prevention and Control of Infectious Diseases, the Indonesian Ministry of Health (MoH). Community Health Centers (*Puskesmas*) and public hospitals collect and report the data to district and province health authorities. We obtained 11 years of monthly dengue case data covering the period 2010–2020 from 514 regencies/cities. From 2010 to 2014, 17 regencies underwent expansion and were, therefore, not included in the analysis because of data limitations. The population and coverage area data (km^2^) at the regency/city level period 2010–2020 was derived from the Indonesian Statistics Bureau (BPS) “https://www.bps.go.id (accessed on 9 August 2022)”. Administrative boundaries on the regency/city level were set using data provided by the Indonesian Geospatial Information Agency (BIG). 

Dengue case data in each regency/city shows varying degrees of completeness with missing data in a number of locations, as described in Figure 1a. We found that 472 regencies/cities have 0–10% missing data, including 263 regencies/cities with 0% missing data and 209 regencies/cities with less than 10% missing data. Additionally, 22 regencies/cities have 10–20% missing data, while no locations have 20–30% missing data. However, 10 regencies/cities have 30–40% missing data, and 10 more 40–50% missing data. The regions with more than 30% missing data are primarily found in newly constituted regencies as a result of the 17 location-expansion zones that we excluded from the analyses. The three other locations with more than 30% missing data are found in locations dominated by zero cases. 

### 2.3. Climate Data

Climate data, consisting of rainfall (RR, mm), relative humidity (RH, %), average temperature (Tavg, °C), minimum temperature (Tmin, °C), and maximum temperature (Tmax, °C), were obtained from the Indonesian Agency for Meteorology, Climatology, and Geophysics (BMKG). Daily climate data was collected from 1 January 2010 to 31 December 2020 from 156 stations. All data from 2010 to 2020 have been analyzed and it was determined that missing values amounted to less than 20% [29,30]. 

As an initial step to ensure data quality, climate data series were flagged for several gross errors, including the absence of a consistent code for rainfall events (e.g., rainfall is recorded as 0 mm or blank entries), duplicate or outlier data, and physical inconsistencies such as a minimum temperature that was the same or higher than the maximum or average temperatures, and vice versa. Manual assessments such as visualizing data and checking data trends were also conducted to double-check data quality. Systematic quality control of the daily climate data was carried out using RClimdex-extraqc “http://www.c3.urv.cat/data/Manual_rclimdex_extraQC.r.pdf (accessed on 31 May 2022)” to detect errors and outlier data and obtain better performance. We compared data from nearby stations to identify suspected outlier data. Expert judgment was also employed if there was no information from nearby stations or other resources to identify outlier data. The monthly climate data scale was used from 128 stations for 131 regencies/cities that have more than 80% data completeness. The spatial distribution of the climate stations in this study is shown in Figure 1b. Additionally, the detailed of climate stations is provided in Appendix A. 

### 2.4. Substituting Missing Data

To tackle the dengue data completeness issue, we used a single imputation method by calculating the mean for an i-th month on specific regencies/cities during the period of data to substitute missing dengue data for the i-th month. This imputation technique considered that dengue cases had a monthly pattern with one peak in several regencies/cities. The mean has been used for some medical data imputation approaches for dengue prediction with a single value in instances with fewer than 15% missing values [31]. Mean imputation can be retained in full datasets and used to run analyses as if data were complete, even though this method does not account for the uncertainty in missing data [32]. 

Missing climate data due to errors in observation instruments, instrument replacements, or human errors can be substituted using spatial interpolation [29]. In this study, we employed the Inverse Distance Weighted (IDW) interpolation method to substitute missing daily climate data. IDW is a widely used deterministic interpolation technique for climate- and weather-related factors that assumes spatial autocorrelation and similarity among points [33]. It operates on the principle that closer points significantly influence interpolation, with diminishing impact as distance increases [34]. In IDW, a linear combination of neighboring observations determines the value at each interpolated location. The details of IDW interpolation can be found in [35,36].

### 2.5. Cluster Analyses

Cluster analysis is a statistical technique that categorizes observational factors into distinct groups. In this study, dengue hotspot areas are delineated based on two key metrics: Incidence Rate (IR) and Case Density (CD). The IR represents the number of dengue cases per 100,000 inhabitants, while the CD indicates the number of dengue cases per square kilometer. This study used K-means clustering to categorize dengue hotspot areas based on the annual averages of IR and CD time series data at the regency and city levels. The data series are standardized to a mean value of 0 and a standard deviation of 1 during the clustering process to ensure equitable consideration of all locations. In K-means, objects are selected and partitioned into k distinct classes. The Euclidean distance metric is utilized to determine the proximity of each input object to the centroids. To identify the optimal number of clusters, we employed the elbow method, assessing up to 10 groupings. Optimal cluster numbers are determined by identifying the point on the elbow plot at which a subtle reduction in within-cluster variability is often visualized as a bend in the elbow [37]. Throughout the K-means clustering process, the value of k is iteratively adjusted, and the within-cluster sum of squares is computed and plotted for each k value.

### 2.6. Singular Value Decomposition (SVD)

SVD is a method in mathematics used to investigate the coupled variability between two variables, identifying modes of behavior where variations in the two variables are closely associated [38]. In meteorology, SVD has been used for data decomposition to find spatial patterns of variability and temporal variations and to assess the significance of patterns and relationships for the two fields. In this study, SVD is used to analyze the variability between the mode climate dataset and dengue incidence in space and time, investigate the dependency of both datasets, and determine the dominant climate parameters in influencing dengue. The climate dataset is defined as an X matrix with *n* time and *p* location points and the IR data is defined as a Y matrix with *n* time and *q* location points. The covariance matrix (C_xy_) was calculated using Equation (1), and the SVD process for C_xy_ was performed using Equation (2) as specified by [38].
(1)$$C_{xy}=X^{T}Y$$
(2)$$C_{xy}=ULV^{T}$$

Two orthogonal spatial pattern clusters and matching pairs of singular values, comparable to eigenvalues, were formed from the cross-covariance matrix in SVD. Equation (2), which includes the three main components, was used to calculate the decomposition of the matrix C_xy_ [38]. The initial values of the fundamental matrix axis were represented by the singular vectors that made up the matrix U, which had a dimension of p × m. The singular value denoting each particular vector relevance within the original matrix was represented by Matrix L. The initial matrix impacted matrix V with a dimension of m × q and was influenced by the original matrix. SVD identifies a linear combination of p predictor variables that exhibits the highest covariance with a linear combination of q predicted variables. These pairs of linear combinations, denoted by A and B, are referred to as expansion coefficients and are analogous to eigenvectors as described in Equation (3).
(3)A=XU and B=YV

A and B contain expansion coefficients for each mode, and due to the orthogonality of U and V, the data matrix can be formed as X = AU^T^ and Y = BV^T^. This provides a straightforward method for evaluating the significance of individual singular modes by considering the proportion of squared covariance, known as the Squared Covariance Fraction (SCF) explained by each mode. SCF expresses the fraction of squared covariance elucidated by the corresponding singular vector. The SVD technique can produce both a homogenous and heterogeneous correlation map. In this study, we used the heterogeneous correlation between the mode of the climate dataset and the IR dataset as another field. 

## 3. Results

### 3.1. Epidemiology of Dengue Cases

Dengue cases were found in most regencies/cities in Indonesia in every year under study. A total of 1,240,267 dengue cases were reported from 2010 to 2020. The highest country-wide counts were in 2010 (156,053) and 2016 (203,936), and the lowest were in 2011 (65,717), 2017 (68,396), and 2018 (65,600). Based on the National Strategic Plan of the Indonesia Ministry of Health for 2020–2024, the IR threshold to control dengue in Indonesia was determined to be less than 49 per 100,000 inhabitants annually. The yearly IR on the regency/city level from 2010 to 2020 is shown in Figure 2. During this time, a high incidence rate (49 ≤ IR ≤ 100) and a very high incidence rate (IR > 100) of dengue were found almost every year, e.g., in East Kalimantan, North Kalimantan, and Bali. In 2016, locations with high and very high IR showed a rapid increase in 250 regencies/cities. Furthermore, 129 regencies/cities with an IR of 49 > IR > 100 and 96 regencies/cities with an IR > 100 were found. 

### 3.2. Determination of the Dengue Hotspot Areas

The K-means clustering technique resulted in four clusters of dengue hotspot areas across Indonesia with representative insight and distinct characteristics based on the dengue IR and CD, as shown in Figure 3. The features of clusters 1 and 2 showed them to be the locations with the most significant dengue incidence. Eleven urbanized sites (2%) with a condensed geographic size and a large number of cases formed cluster 1. Twenty-five locations (5%), mainly in districts with sizable areas and many dengue cases, formed cluster 2. Additionally, cluster 3 had 117 locations (24%) with a medium level of dengue incidence and case density, whereas cluster 4 contained 344 regencies/cities (69%) with a low level of dengue incidence and case density. From the clustering result, we were able to determine locations to prioritize for the management and control of the incidence of dengue. Cluster 1 and cluster 2 were classified as areas where there is a high priority for dengue control due to increased dengue cases almost every year. However, the control programs for dengue in cluster 1 and cluster 2 should be different because cluster 1 has a smaller area than cluster 2. Cluster 3 and cluster 4 were identified as areas where there is a medium and low priority for dengue control, respectively. In cluster 3, dengue incidence was frequently found every year, but with a medium number of cases, and in cluster 4, a low number of cases were found. The detail of cluster number in each province and the cluster member is provided in Appendix A, respectively. 

The spatial distribution of the clustering result is shown in Figure 4, (a) for all of Indonesia, and (b) for Java, Bali, and Lombok Island. Red denotes cluster 1, and orange, green, and yellow denote cluster 2, cluster 3, and cluster 4, respectively. Cluster 1 is dominantly located in Java island, Bali, and Lombok as visualized in Figure 4b. The identified dengue hotspot areas can be utilized to analyze spatial and temporal patterns of dengue incidence and their correlation with dominantly climatic factors in Indonesia. Maps depicting the hotspot dengue areas in Indonesia can also be used to identify priority areas for dengue control efforts. One of the measures to control dengue spread involves devising preventive steps by developing early warning systems for dengue and utilizing climatic factors as input parameters for dengue prediction models.

### 3.3. Spatiotemporal Distribution of Climate Parameters Strongly Correlated with Dengue Incidence

#### 3.3.1. Variance and Coefficient Correlation

The dominant patterns in a time series of climate parameters and IR were obtained using the SVD technique. Each of the modes extracted will be orthogonal to the others. The SVD produced covariance (SCF) and coefficient correlation (r) values that showed strong, significant coupling between climate parameters and IR. The first mode (mode 1) will be the most coupled pattern and exhibit a highly significant variance. Subsequently, the second mode emerges as the second most dominant, followed by the third mode, and so on. The SVD results show the dependency of dengue cases or incidence on climate parameters across Indonesia. The number of regions/cities utilized in applying the SVD technique aligns with the availability of climate stations, totaling 131 regions/cities. Eight districts/cities were used in Cluster 1, 11 in Cluster 2, 39 in Cluster 3, and 73 in Cluster 4. The two modes of SCF values and the correlation between expansion factor A (the first mode of climate) and expansion factor B (the first mode of IR) are shown in Table 1. 

The variance values expressed as SCF (%) show that mode 1 and mode 2 of the climate predictors can explain most of the diversity in predicting IR. As we move to mode 3 and beyond, the SCF values decrease continuously. Based on Table 1, the total SCF exceeds 80% for all clusters and climate parameters from the first mode to the second. Relative humidity (RH) achieved the highest covariance values in the first mode of cluster 1, cluster 2, cluster 3, and cluster 4, with percentages of 98.7%, 73%, 90%, and 86%, respectively (see Table 1). The correlation values (r) between the expansion factors of climate parameters and IR indicate the relationship strength between these two parameters. These correlation values vary among the first and second modes in each cluster. The highest correlation in the first mode for cluster 1 is found with air humidity (0.47), in cluster 2, it is found with rainfall (0.54) and average temperature (0.54), in cluster 3, with rainfall (0.57) and average temperature (0.57), and in cluster 4, with rainfall (0.58). The 95% significance level was used for the first and second mode of climate parameters in all clusters that showed significance with *p*-values less than 0.05 (Table 1). 

#### 3.3.2. Time-Series Patterns of the Expansion Factors 

The time-series patterns between the first mode of each climate parameter and the first mode of the IR on a monthly scale are shown in Figure 5. Figure 5a–d show the time series pattern for cluster 1, cluster 2, cluster 3, and cluster 4, respectively. In Figure 5a, the IR shows clear monthly fluctuations with a single peak around February until April, generally following the pattern of each climate parameter. Increases in rainfall and humidity correspond to increases in IR. The IR peaks notably occurred in 2010 and 2016 within cluster 1. A decline in IR was also noticeable in 2017 and 2018 following the surge in cases in 2016. Figure 5b shows the temporal pattern of IR concerning rainfall (a) and air humidity (b) in cluster 2, exhibiting clear seasonal patterns, while temperature shows a different pattern concerning IR. In cluster 3, as depicted in Figure 5c, the temporal pattern of IR concerning rainfall and humidity displays a distinct seasonal pattern with a single peak, whereas the temperature parameter shows an inverse pattern. In Figure 5d, the highest IR peaks in cluster 4 are generally observed in 2016 and 2019. The monthly IR pattern shows relatively lower fluctuations but still generally follows the climate pattern.

#### 3.3.3. Spatial Distribution of Dominant Climate Parameters to Dengue Incidence

The SVD technique resulted in the heterogeneous correlation between the expansion factors of the mode of each climate parameter and dengue incidence. For each regency/city, we chose the parameter with the highest correlation among the first modes of RR, RH, Tavg, Tmax, and Tmin toward IR (A1Y). The spatial map is obtained from the combination of heterogeneous correlation in all clusters used to determine the most dominant climate parameters with the highest magnitude and specify the appropriate location. The spatial distribution of heterogeneous correlation values between the first mode of climate parameters and IR for all clusters is depicted in Figure 6. 

Based on Figure 6a,b, the map showed that the heterogeneous correlations between the first mode of rainfall (RR) and the first mode of relative humidity (RH) towards IR have a higher magnitude than that of the first mode of temperature, ranging from 0.30 to 0.56. These correlation values exceeded 0.30 in a total of 52 regencies/cities for relative humidity (RH), followed by 49, 17, 29, and 2 regencies/cities for RR, Tavg, Tmax, and Tmin respectively. The predominant spatial distribution of correlation values between RH and IR, as well as between RR and IR, was found in the southern part of Indonesia, i.e., Java, Bali, Nusa Tenggara, the eastern part of Sumatra, the southern part of Kalimantan, and several regencies/cities in the southeastern and northern regions of Sulawesi. Conversely, correlation values below −0.30 were identified in 19 regencies/cities for Tavg, 19 for Tmax, and 31 for Tmin.

## 4. Discussion

### 4.1. Dengue Hotspot Areas

Dengue has been spreading rapidly to almost all regencies/cities and is frequently found every year in Indonesia. From 2010 to 2020, a total of 1,240,267 dengue cases were reported nationally. East and North Kalimantan, Bali, and DKI Jakarta were the provinces that frequently had a high number of dengue cases. In the last five decades, several peaks in IR have been identified in Indonesia with a cyclic pattern that peaked approximately every 6 to 8 years, as in 1973, 1988, 1998, 2009, and 2016 [3]. The change in the trend in IR during the 2000s peaked in 2016, partly due to a shift in the dominant serotypes from DEN-3 to DEN-1 and DEN-2, which rapidly emerged in Indonesia. Additionally, there was heightened activity of these virus serotypes across most areas in Indonesia [3]. For instance, during the dengue fever outbreak in Jambi in 2015, DEN-1 dominated at an incidence rate of 66%, causing more severe clinical impacts than DEN-3 [39]. This change in serotype dominance was also observed regionally in other regions, such as Southeast Asia [3].

The most important result of the clustering technique using IR and CD was a categorization of dengue hotspot areas as priority regions for intervention and control of dengue. The first category is a high–interest area, consisting of cluster 1 and cluster 2, which had a high number of dengue cases but different case densities. Cluster 1 was dominated by urban areas, commonly found in relatively confined areas with high population densities. Conversely, cluster 2 is located in urban and suburban areas with more expansive areas than cluster 1. In densely populated and narrow urban areas, the probability of human and vector contact is very high [40], which could trigger fast dengue transmission. *Aedes aegypti* is also often found in houses in urban areas [41], so the denser the settlement, the easier mosquitoes circulate and the faster the virus transmits. In urban areas, there are also many people with no immunity to one of the four DEN serotypes for which an effective dengue vaccine is still unavailable. An explosive dengue outbreak will occur if climate conditions favor dengue vector and virus spread [40]. In suburban areas, which commonly have larger areas and lower population densities than urban areas, many dengue cases were also found. Cluster 3 was identified as a medium interest area for the dengue control program. In this cluster, dengue cases are frequently found annually, with an incidence rate reaching a high government threshold (IR > 49) with case density similar to cluster 2. The last category is the low-interest area, which consists of cluster 4. In this cluster, dengue cases were rarely found annually; if cases were found, there were few in number. The identified dengue hotspot areas with different degrees of dengue incidence can be used by health authorities like the Ministry of Health and local governments to determine the regency/city prioritization for the prevention and control of dengue transmission effectively. 

### 4.2. Climate Parameters Dominant in Influencing Dengue Incidence

Rainfall, humidity, and temperature directly impact the mosquito life cycle at every stage and can influence dengue transmission indirectly. Additionally, these climate factors contribute to a heightened probability of human–mosquito vector contact, consequently, elevating the transmission of the dengue virus [5]. Each climate parameter influences dengue incidence to a different degree in each area [22,23,24,42]. Determining which climate parameters dominantly influence dengue incidence in Indonesia and the dependency between dengue incidence and climate parameters is essential. Based on the SVD results, the heterogeneous correlation between the first mode of each climate parameter and IR shows that rainfall and relative humidity have higher degrees of correlation than average, maximum, and minimum temperatures. These correlation values reached more than 0.3 and were primarily found in the southern part of Indonesia with the monsoonal rainfall pattern. In the monsoonal climate region, the rainfall is influenced by the Asian winter monsoon, which brings more water vapor from the South China Sea, generally from November to March (wet season). Conversely, from May to September, the monsoonal wind will bring drier air from Australia (dry season) [27]. During the dry season, mosquito eggs will survive and hibernate in their breeding places for months. When the rainy season begins and with the watering of the storage containers, the eggs that hibernate during the dry season will soon hatch. In relation to this hatching and the growth to adulthood, the pattern of dengue cases generally followed the rainfall with a lag time around of two months [22]. This pattern shows that the mosquito egg will start to hatch and the mosquitoes grow to adulthood when the rains fall at the onset of the wet season. We also obtained from Figure 6a that a very weak correlation between the first mode of rainfall and IR (r ≤ 0.1) was found in several locations with an equatorial rainfall pattern. This type of rainfall has two peaks in the year, generally in October to November and March to May [27]. A better correlation between rainfall and dengue incidence in this rainfall pattern defined the second mode. 

Relative humidity and temperature provide a favorable environment for mosquitoes to lay eggs and survive until they hatch. The optimal relative humidity was found to be more than 75% for egg preservation and also has a positive correlation with egg-hatching ability [43]. This factor was also observed to significantly impact the vector’s ability to transmit a pathogen, behavior of biting, and adult mosquito survival rate [2]. In tropical areas surrounded primarily by oceans, the temperature is relatively warm throughout the year, providing a sufficient environment for mosquitoes to continue laying eggs. With temperatures ranging from 23 °C to 31.6 °C, the lowland areas over Indonesia [28] are suitable for mosquito life and transmission of the dengue virus to humans [22]. The heterogeneous correlation between the first mode of average temperature and incidence rate (Figure 6c), as well as maximum temperature and incidence rate (Figure 6d), indicated a scattered distribution in all locations, exhibiting both positive and negative correlations. For the minimum temperature (Figure 6e), a negative correlation with a magnitude ranging from −0.2 to −0.45 was found in almost all locations. The vector development will be hampered by extreme temperatures that are too low or too high. In specific environments, higher temperatures can consequently, raise mosquito mortality rates and reduce the risk of dengue transmission [2]. Moreover, the results also showed that in highland areas, a negative correlation with the highest magnitude between the first mode of minimum temperature and incidence rate was defined in Kerinci regency (elevation ±784 m), Malang (±590 m), Tana Toraja (±829 m), and Kepahiang (±517 m). In these highland areas, the dengue case data is less than 10 cases in a month and these areas are mostly classified in cluster 3 and cluster 4. In other highland areas, a positive correlation was found between the first mode of relative humidity and incidence rate in Kota Bandung (±791 m) in cluster 1, Pasuruan (±832 m) in cluster 3, and Tana Toraja in cluster 4. 

In influencing vector growth and virus transmission, climate variables should not be considered independently, as they support each other in influencing dengue transmission. Elevated rates of precipitation, coupled with higher temperatures, also lead to increased humidity [44]. The results of this study provide valuable insight into how far the climate influences dengue in wide areas of Indonesia. This result also can be used as a preliminary reference in building climate–dengue model predictions regarding prioritized dengue hotspot areas. Even if the climate–dengue prediction cannot easily be modeled because of their non-linear relationships, it can be utilized for a dengue–climate early warning system. It represents the first piece of information that should be used to enhance awareness and determine preventive strategies for dengue outbreaks. Further strategies can be employed by health authorities to minimize the risk of dengue transmission, such as vector control, sanitation improvement, public education, and environmental management.

This study had several limitations. Dengue case data at the national level were only partially available; some instances of missing data were encountered and data were available only at the regency/city level. The data did not differentiate between the three types of dengue, i.e., Dengue Fever, Dengue Hemorrhagic Fever, and Dengue Shock Syndrome. Information is also needed indicating whether the data has undergone epidemiological investigation or not. The second limitation is that the use of observation data from climate stations was not enough to cover all regencies/cities in Indonesia. The distribution of climate stations is still sparse and only covers around 25% of the country’s 514 regions/cities. Gridding climate data from a regional climate model or reanalyzing climate data to cover all regencies/cities in Indonesia is recommended. Further improvement is needed, such as using lag-time and involving non-climatic factors in the analyses.

## 5. Conclusions

Dengue hotspot areas in Indonesia’s regions have been identified based on dengue incidence rate and case density at the regency/city level. Cluster 1 predominantly consists of cities and is characterized by moderate to high incidence and high case density. Cluster 2 encompasses areas with high incidence but lower case density. Both cluster 1 and cluster 2 represent areas where there is a high priority for intervention and control of dengue. Cluster 3 includes regions with moderate incidence and case density, categorized as medium priority. Lastly, cluster 4, characterized by low incidence and case density, falls into the low-priority category.

The application of the SVD technique revealed that the highest covariance values were associated with relative humidity in the first mode of climate parameters and IR across all clusters. In this first mode, cluster 1 exhibits the highest correlation values with relative humidity, while cluster 2 and cluster 3, the highest correlation values were with rainfall and average temperature, and cluster 4 showed higher correlations with rainfall. The heterogeneous correlation showed that the dominant temporal pattern for IR across all clusters follows a distinct pattern, aligning with the first mode of rainfall and relative humidity patterns. However, the temperature exhibits a different pattern. In the spatial distribution of correlation values, the first mode of rainfall and relative humidity were the dominant climate parameters in influencing the incidence of dengue in Indonesia, with a higher correlation than temperature. This dominant correlation was found in Java, Bali, Nusa Tenggara, eastern Sumatra, southern Kalimantan, and several locations in the northern and southeastern parts of Sulawesi. 

Determining the dengue hotspot areas is crucial for directing resources and efforts toward controlling dengue outbreaks effectively. Additionally, our study underscores the pivotal role of certain climate parameters in influencing dengue transmission, offering valuable insights for developing dengue–climate model prediction. Such models can empower health authorities to implement proactive measures to prevent dengue transmission and protect public health. 

## Figures and Tables

**Figure 1 insects-15-00366-f001:**
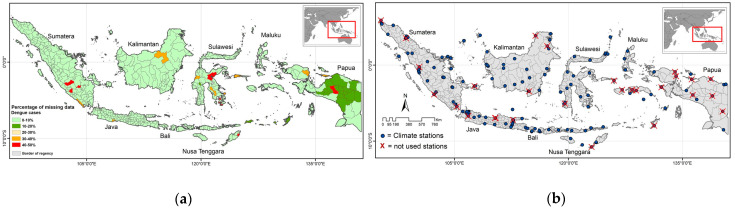
Spatial distribution of data completeness in Indonesia: (**a**) for dengue data, (**b**) for 156 climate stations. Stations marked with blue dot represent those stations used in analyses, while marked X represent those stations excluded from analyses because of a lack of data completeness.

**Figure 2 insects-15-00366-f002:**
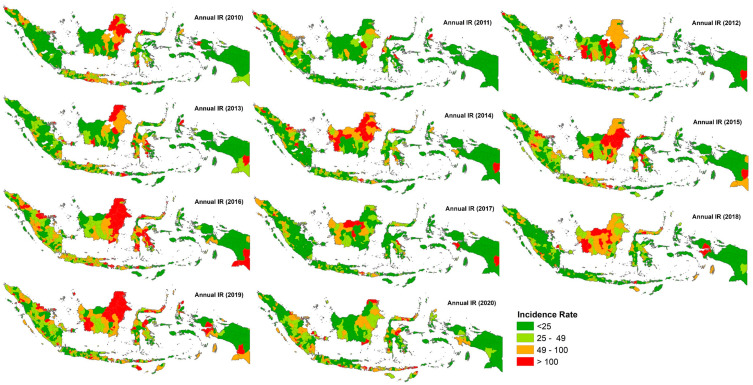
Geographical distribution of annual incidence rate (cases per 100,000 inhabitants) from 2010 to 2020 in Indonesia.

**Figure 3 insects-15-00366-f003:**
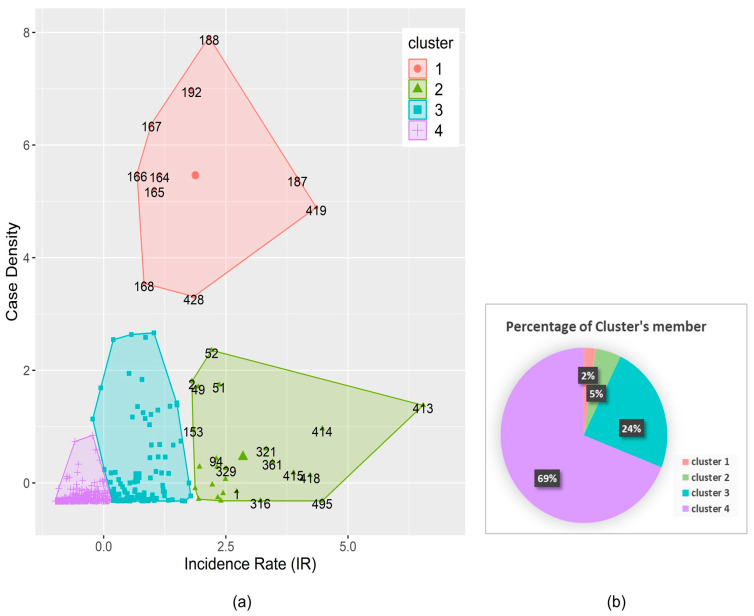
(**a**) Four clusters of dengue endemic regency/city level, and (**b**) the percentage of cluster members based on incidence rate and case density for the period 2010–2020.

**Figure 4 insects-15-00366-f004:**
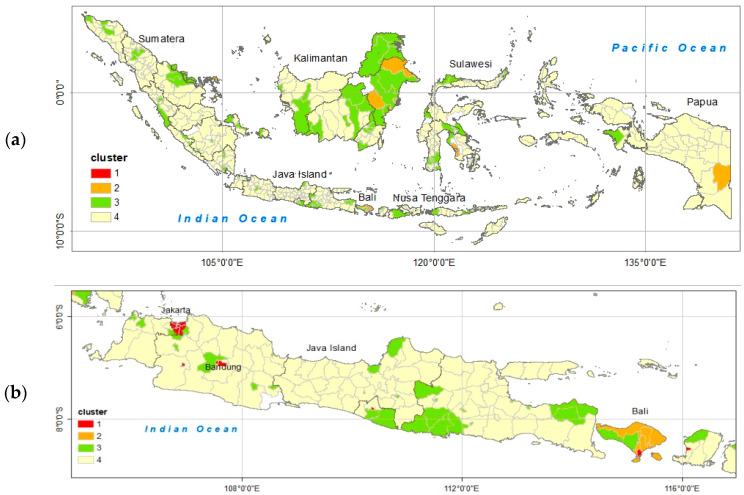
Map of dengue hotspot areas based on clustering: (**a**) national level, (**b**) Java Island, Bali, and Lombok.

**Figure 5 insects-15-00366-f005:**
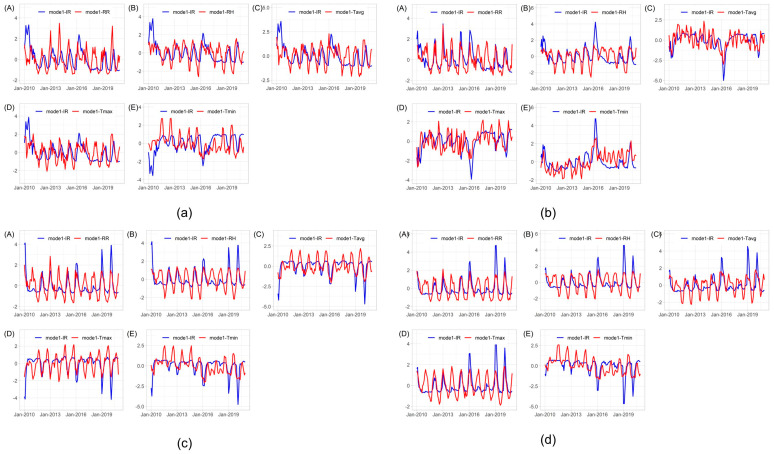
Temporal patterns of coefficient expansion of the first mode of climate factors and dengue incidence rate for each cluster; (**a**) cluster 1, (**b**) cluster 2, (**c**) cluster 3, and (**d**) cluster 4, and for climate parameters for each cluster; (**A**) RR, (**B**) RH, (**C**) Tavg, (**D**) Tmax, and (**E**) Tmin.

**Figure 6 insects-15-00366-f006:**
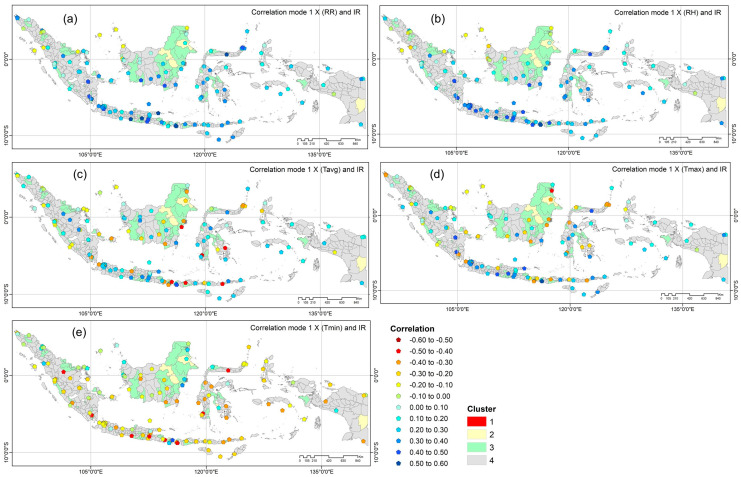
The heterogeneous correlation map between the expansion coefficients of the first mode of climate factors and IR in all clusters; (**a**) RR, (**b**) RH, (**c**) Tavg, (**d**) Tmax, and (**e**) Tmin.

**Table 1 insects-15-00366-t001:** The SCF value and correlation between expansion coefficients A and B (two orthogonal spatial sets) of the two first modes of each cluster.

Cluster	Mode	SCF (%)					Correlation (r) *				
		RR	RH	Tavg	Tmax	Tmin	RR	RH	Tavg	Tmax	Tmin
Cluster 1	Mode 1	95.0	98.7	88.3	92.7	79.4	0.24 (0.005)	0.47 (0.00)	0.33 (0.00)	0.46 (0.00)	0.24 (0.005)
	Mode 2	3.8	0.9	9.5	4.0	12.0	0.19 (0.026)	0.40 (0.00)	0.31 (0.00)	0.23 (0.007)	0.20 (0.022)
Accum. SCF mode 1–2		98.8	99.6	97.8	96.7	91.4					
Cluster 2	Mode 1	69.6	73.2	69.5	56.6	79.6	0.54 (0.00)	0.49 (0.00)	0.54 (0.00)	0.51 (0.00)	0.49 (0.00)
	Mode 2	15.3	13.4	20.8	28.1	15.2	0.39 (0.00)	0.49 (0.00)	0.42 (0.00)	0.53 (0.00)	0.45 (0.00)
Accum. SCF mode 1–2		84.9	86.6	90.3	84.7	94.8					
Cluster 3	Mode 1	84.0	90.3	59.6	79.6	80.5	0.57 (0.00)	0.54 (0.00)	0.57 (0.00)	0.54 (0.00)	0.48 (0.00)
	Mode 2	5.8	4.2	26.2	8.7	12.9	0.47 (0.00)	0.59 (0.00)	0.60 (0.00)	0.49 (0.00)	0.65 (0.00)
Accum. SCF mode 1–2		89.8	94.5	85.8	88.3	93.4					
Cluster 4	Mode 1	82.3	86.1	69.1	69.0	83.9	0.58 (0.00)	0.53 (0.00)	0.45 (0.00)	0.54 (0.00)	0.49 (0.00)
	Mode 2	4.0	5.1	13.3	14.7	7.5	0.53 (0.00)	0.56 (0.00)	0.46 (0.00)	0.50 (0.00)	0.63 (0.00)
Accum. SCF mode 1–2		86.3	91.2	82.5	83.7	91.4					

* In parenthesis are the 95% significance levels for the correlation (α < 0.05).

## Data Availability

The authors do not have authorization to share the dataset. The climate data can be accessed from data online “https://dataonline.bmkg.go.id/home (accessed on 26 October 2021)”.

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
