# Peer review of "Spatiotemporal Characterization of Dengue Incidence and Its Correlation to Climate Parameters in Indonesia"

_insects, 2024, doi:10.3390/insects15050366_

Round 1

Reviewer 1 Report

Comments and Suggestions for Authors

Comments on: Spatiotemporal characterization of dengue incidence and its correlation to climate parameters in Indonesia

This manuscript explored variations in time of dengue incidence across Indonesia to understand the effect of several weather parameters. The results showed stratification into four clusters with varying degrees of disease incidence, related to dominant weather parameters, mainly rainfall and relative humidity.

The introduction does not seem to include previous studies where spatial stratification was conducted with similar objectives to this study or where this type of study has been used to improve the control of dengue. Recommendations coming out of this study are lacking in terms of improving dengue control.

It called my attention that the authors did not include important variables related to dengue transmission, such as human population density, elevation above sea level, etc. Thus, their analysis of a few highly correlated variables (rainfall, relative humidity; average, minimum, maximum temperature) is overly simplistic and possibly redundant.

The manuscript needs a thorough revision of grammar. It is difficult to read and full of technical words related to the statistics used, that most readers would lose interest very soon. The use of plain language and substantially reducing the size of the manuscript is recommended.

Specific comments are included in the pdf file.

Comments on the Quality of English Language

Needs extensive improvement in grammar.

Author Response

Dear Reviewer,

Hereby, attached our response and some corected statements.

Thankyou.

Reviewer 2 Report

Comments and Suggestions for Authors

There was a lot of work put into the manuscript especially with the nice figures.  However, there are a number of grammatical issues with the paper.  subject verb agreement is an issue.  Abbreviate the species name after the first time mentioned in the manuscript.  Should avoid using apostrophes in scientific literature.  The introduction needs to be condensed.  Start with the broad picture then narrow in on the specific intentions of the research.  

I do think the use of endemic is wrong in the manuscript.  The authors mention This study was designed to determine the dengue endemic area based on the IR and CD of dengue (Lines 476 to 477)... I believe what the findings are presenting are hot spot areas of dengue IR and CD that should be targeted for interventions. 

The manuscript needs major editing.  

Comments on the Quality of English Language

The manuscript needs major editing.  

Author Response

Dear Reviewer,

Hereby, attached our revisions and some corected statements, according to your reviewed comments.

Thankyou.

Round 2

Reviewer 1 Report

Comments and Suggestions for Authors

The authors addressed the comments and suggestions